# Isolation, Characterization, Genome Analysis and Host Resistance Development of Two Novel *Lastavirus* Phages Active against Pandrug-Resistant *Klebsiella pneumoniae*

**DOI:** 10.3390/v15030628

**Published:** 2023-02-25

**Authors:** Mina Obradović, Milka Malešević, Mariagrazia Di Luca, Dušan Kekić, Ina Gajić, Olivia McAuliffe, Horst Neve, Nemanja Stanisavljević, Goran Vukotić, Milan Kojić

**Affiliations:** 1Laboratory for Molecular Microbiology, Institute of Molecular Genetics and Genetic Engineering, University of Belgrade, 11000 Belgrade, Serbia; 2Department of Biology, University of Pisa, 56127 Pisa, Italy; 3Institute of Microbiology and Immunology, Faculty of Medicine, University of Belgrade, 11000 Belgrade, Serbia; 4Department of Food Biosciences, Teagasc Food Research Centre, P61 C996 Fermoy, Ireland; 5Department of Microbiology and Biotechnology, Max Rubner-Institut, 24103 Kiel, Germany; 6Faculty of Biology, University of Belgrade, 11000 Belgrade, Serbia

**Keywords:** *Lastavirus*, lytic bacteriophage, *Klebsiella pneumoniae*, tail fiber, phage resistance

## Abstract

*Klebsiella pneumoniae* is a global health threat and bacteriophages are a potential solution in combating pandrug-resistant *K. pneumoniae* infections. Two lytic phages, LASTA and SJM3, active against several pandrug-resistant, nosocomial strains of *K. pneumoniae* were isolated and characterized. Their host range is narrow and latent period is particularly long; however, their lysogenic nature was refuted using both bioinformatic and experimental approaches. Genome sequence analysis clustered them with only two other phages into the new genus *Lastavirus*. Genomes of LASTA and SJM3 differ in only 13 base pairs, mainly located in tail fiber genes. Individual phages, as well as their cocktail, demonstrated significant bacterial reduction capacity in a time-dependent manner, yielding up to 4 log reduction against planktonic, and up to 2.59 log on biofilm-embedded, cells. Bacteria emerging from the contact with the phages developed resistance and achieved numbers comparable to the growth control after 24 h. The resistance to the phage seems to be of a transient nature and varies significantly between the two phages, as resistance to LASTA remained constant while resensitization to SJM3 was more prominent. Albeit with very few differences, SJM3 performed better than LASTA overall; however, more investigation is needed in order to consider them for therapeutic application.

## 1. Introduction

*Klebsiella pneumoniae* is a member of the perilous ESKAPE group of bacteria (*Enterococcus faecium, Staphylococcus aureus, Klebsiella pneumoniae, Acinetobacter baumannii, Pseudomonas aeruginosa, Enterobacter* spp.) that threaten even the best healthcare systems in the world. In a recent comprehensive analysis of the global impact of antimicrobial resistance (AMR), *K. pneumoniae* was found to be one of the three major contributors in both analyzed categories: deaths attributable to AMR and deaths associated with AMR, alongside *E. coli* and *S. aureus* [1]. It is a Gram-negative, encapsulated and non-motile opportunistic pathogen present in the environment, in soil and in wastewaters, but is also a benign colonizer of human mucosa [2]. Acquisition of genetic elements for antibiotic resistance and the emergence of hypervirulent strains, defined as strains that have the ability to infect healthy people aside from the immunocompromised and cause a wider variety of health complications than classic strains, escalate the severity of *K. pneumoniae* infections, causing urinary tract infections, pneumonia, bacteremia or liver abscesses [3].

The use of polymyxins, such as colistin, to combat carbapenemase-producing *K. pneumoniae* strains resulted in outbreaks of colistin-resistant strains, reported in many countries, including Serbia [4,5,6,7,8,9]. With the concerning increase of antibiotic resistance and hypervirulence in nosocomial *K. pneumoniae* strains, there is an urgent need for alternative and more effective treatment approaches. Bacterial viruses or bacteriophages are being evaluated as a potential way to combat multidrug-resistant pathogens. Due to host range specificity, bacteriophages are suitable for precisely targeting the infection-causing bacteria. In order to be applied in therapy, it is required that phages meet certain criteria such as being obligately lytic or lacking lysogenic traits, transduction potential and genes encoding for toxins or antibiotic resistance, and preferably have a broad host range [10]. Several studies of phage therapy against *K. pneumoniae* infections have been conducted, both on mouse models [11] or as clinical interventions [12,13]. A case report by Cano et al. [12] puts an optimistic perspective on the future of phage therapy and treating medical devices and biofilm-related infections. In the mentioned study, intravenous application of a single bacteriophage (along with an antibiotic) resolved a complicated limb-threatening *K. pneumoniae* infection. Successful implementation of phage therapy in the treatment of pandrug-resistant *K. pneumoniae* infection has also been reported recently by Eskenazi et al. [13]. After long-term antibiotic therapy for a fracture-related infection, preadapted phages were locally administered to the wound, which resulted in the improvement of the patient’s overall condition.

One of the major limitations of phage therapy is phage resistance. Studying how consistently phage resistance evolves in vitro, Hesse and co-workers [14] observed that mutations developed in the *K. pneumoniae* genome in the presence of phages are most commonly found in genes related to the synthesis or assembly of cell surface structures, resulting in impaired phage adsorption. On the other hand, a study on the *E. coli* population observed that bacteriophages co-evolve with their hosts, shaping their tail fibers for better recognition and attachment to the bacterial cells [15]. By conducting an *in vitro* evolution experiment, de Leeuw et al. [16], studying *Aquamicrobium* H14 and *Alcaligenaceae* H5 strains, concluded that the most commonly occurring mutations in a phage’s genome appeared in tail-related regions.

Capsular polysaccharide (CPS) is the most thoroughly studied and the most important virulence factor of *K. pneumoniae*. Based on the K antigen (capsule polysaccharide), at least 79 capsular serotypes have been reported for *K. pneumoniae* [17]. Hypervirulent (HV) serotypes (mostly/predominantly K1 and K2) have increased production of CPS and create a mucoviscous hypercapsule that increases the degree of pathogenicity, allowing HV strains to establish invasive infections in previously healthy persons [18,19]. Upon meeting capsule-targeting phages, the evolution of *K. pneumoniae* strains might lead to loss or modification of the capsule, discarding of multidrug resistance clusters or resensitization to phages which use different receptors [20].

In our study, we have isolated two bacteriophages of *K. pneumoniae*, vB_KpnP_LASTA and vB_KpnP_SJM3, hereafter abbreviated as LASTA and SJM3, respectively. Detailed characterization and bioinformatic analyses have been performed. The possibilities of integrating phage genomes into the host have been tested, as well as the phages’ lytic and antibiofilm activity. The occurrence of host strain phage-resistant mutants and their resensitization was also monitored.

## 2. Materials and Methods

### 2.1. Bacterial Strains and Culture Conditions

In this study, 140 *K. pneumoniae* isolates of clinical origin were assayed for phage susceptibility as described below. Twenty-seven of these belonged to the Laboratory for Molecular Microbiology, Institute of Molecular Genetics and Genetic Engineering, University of Belgrade and were characterized in our previous work [4]. The remaining 113 bacterial isolates were obtained from the collection of the Institute of Microbiology and Immunology, Faculty of Medicine, University of Belgrade. They originated from different clinical laboratories of secondary and tertiary hospitals throughout Serbia. The isolates were obtained from adult patients with confirmed various intrahospital infections between 2016 and 2019. Routine microbiological identification was done using Vitek 2, MALDI-TOF and API 20E. Antimicrobial susceptibility testing was done according to EUCAST standards for the respective year. The *K. pneumoniae* Ni9 strain [4] was further used as a host bacterium for bacteriophage isolation and propagation.

All bacterial strains were propagated in laboratory-made Luria–Bertani (LB) medium (10 g tryptone, 5 g yeast extract, 5 g NaCl, 1 L water), at 37 °C with aeration of 180 rpm. LA (LB with addition of 1.5% agar) was used as solid medium for bacterial propagation. The same media was used for phage propagation and testing. Serial dilutions of bacteriophages were prepared in laboratory-made SM buffer (100 mM NaCl, 8 mM MgCl_2_ × 7H_2_O, 50 mM Tris-HCl pH 7.5). The titer of phages was determined with double agar overlay plaque assay [21], by mixing 100 µL of phage dilution and 15 µL of overnight culture (2 × 10^9^ CFU/mL) of Ni9 strain in 10 mL of melted LB top agar (0.75% agar) and pouring over an LA layer in Petri dish.

### 2.2. Pulsed-Field Gel Electrophoresis

In order to investigate the similarity between the *K. pneumoniae* strains sensitive to phages LASTA and SJM3, or between resistant offspring and parent Ni9 strains, bacterial DNA was analyzed by pulsed-field gel electrophoresis (PFGE) as previously described by Vukotic et al. [22] with minor modifications. Bacterial genomic DNA was digested with *Xba*I restriction enzyme (Thermo Fisher Scientific, Waltham, MA, USA).

### 2.3. Phage Isolation, Purification and Propagation

Bacteriophages LASTA and SJM3 were isolated from two sewage wastewater samples collected at different locations in Belgrade, Serbia by the standard method of homogeneous enrichment using the *K. pneumoniae* Ni9 strain culture as host bacterium. Phage isolation, enrichment and purification methods used were previously described by Vukotic et al. [22]. Briefly, sewage water filtrate was mixed with 2× concentrated LB medium and an overnight culture (2 × 10^9^ CFU/mL) of host strain Ni9. After 16 h incubation at 37 °C, the collected supernatant of the potential lysate was filtrated (Filtropur S 0.22; Sarstedt, Nümbrecht, Germany) and tested for phage presence by double agar overlay plaque assay [21]. Detected phages were purified from samples through four rounds of single plaque purification [23]. A single plaque from the last round of plaque purification was used for the production of a working phage suspension, used in further tests. Phage culture purification and concentration was performed using ultracentrifugation in a gradient of CsCl [23]. A 500 mL amount of mid-exponential-phase Ni9 culture (OD_600_ 0.2, 1 × 10^8^ CFU/mL) was infected with 1 mL of phage suspension (titer 10^9^ PFU/mL) and incubated overnight at 37 °C on a rotary shaker (180 rpm). The supernatant of the lysate was collected after centrifugation and treated with DNase I and RNase A (Thermo Fisher Scientific, Waltham, MA, USA) for 2 h at 37 °C, followed by phage precipitation with the addition of 1 M NaCl and 10% polyethylene glycol (PEG 8000; Sigma) at 4 °C overnight. Pelleted precipitate was resuspended in SM buffer, and CsCl was added at a concentration of 0.75 g/mL. Samples were ultracentrifuged at 131,980.8× *g* for 24 h in a Beckman Coulter Optima L-80 XP ultracentrifuge in rotor SW55 Ti. A distinct pale blue band containing phages was extracted from the gradient using a syringe and 22-gauge needle (PrecisionGlide™; Becton Dickinson, Dublin, Ireland), further dialyzed overnight against 2 L of SM buffer at 4 °C and, finally, filtered by passing through 0.22 µm filters (Filtropur S 0.22; Sarstedt, Nümbrecht, Germany). In this manner, concentrated and purified final phage suspension was used in subsequent experiments.

### 2.4. Transmission Electron Microscopy of Phage Particles

Phage cultures purified by CsCl gradient centrifugation were dialyzed against phage buffer (20 mM Tris-HCl pH 7.2, 10 mM NaCl, 20 mM MgSO4) by 20 min microdialysis on 0.025 μm VSWP membrane filters (Merck Millipore, Darmstadt, Germany). Ultrathin carbon films (~3 × 3 mm in size) were floated from mica-sheets into a drop of phage culture (100 μL, 2 × 10^10^ PFU/mL), and after an adsorption time of 20 min, samples were transferred into a drop of 1% (vol/vol) of electron-microscopy-grade glutaraldehyde (20 min) for fixation. Negative staining with 2% uranyl acetate and subsequent transmission electron microscopy (TEM) (Tecnai 10; FEI Thermo Fisher Scientific) was done as described earlier [24] at an acceleration voltage of 80 kV. Micrographs were captured with a MegaView G2 charge-coupled device camera (Emsis, Muenster, Germany).

### 2.5. DNA Extraction and Manipulation

DNA from bacteriophage lysates was isolated using Phage DNA Isolation Kit (Norgen Biotek, Ontario, Canada) according to the manufacturer’s instructions. Extraction of bacterial DNA and of DNA from a smaller volume of purified and concentrated phage culture (2 × 10^10^ PFU/mL) was performed using the phenol–chloroform DNA extraction method [25]. The integrity and quantity of isolated DNA was inspected by agarose gel electrophoresis [26] (1% agarose) and visualized under UV light after ethidium bromide (SERVA Electrophoresis GmbH) staining.

### 2.6. Genome Sequencing and Bioinformatic Analysis

Total genomic DNAs of bacteria and phages were sequenced using the Illumina HiSeq 2500 platform by MicrobesNG (MicrobesNG, IMI-School of Biosciences, University of Birmingham, Birmingham, UK). The *de novo* assembly was performed using the De Bruijn graph method and the contigs shorter than 200 bp were eliminated [27]. Raw reads were mapped to assembled scaffolds with the Burrows–Wheeler Aligner, BWA [28]. Gene annotation and prediction of the open reading frames (ORFs) of the whole-genome sequences were conducted by the NCBI Prokaryotic Genome Annotation Pipeline (PGAP, https://www.ncbi.nlm.nih.gov/genome/annotation_prok/, accessed on 22 February 2023) service, accessed on 7 December 2022. Nucleotide sequence alignment of LASTA and SJM3 to other publicly available *K. pneumoniae* phages was done using BLAST (https://blast.ncbi.nlm.nih.gov/Blast.cgi, accessed on 22 February 2023) and DNA Strider software v 2.0 f1.3. The phage proteomic tree of viral genome sequences based on genome-wide sequence similarities computed by tBLASTx was generated by ViPTree, the viral proteomic tree server, version 1.9.1 (https://www.genome.jp/viptree/, accessed on 22 February 2023) [29]. The potential presence of genes associated with virulence and antibiotic resistance in the phage genome was analyzed by VirulenceFinder v. 2.0 (https://cge.cbs.dtu.dk/services/VirulenceFinder/, accessed on 22 February 2023) and ResFinder ver. 4.1 (https://cge.cbs.dtu.dk/services/ResFinder/, accessed on 22 February 2023) [30]. Genomes were searched for Rho-factor-independent terminators with ARNold software [31], and promoter sequences were identified with PHIRE [32].

Whole-genome sequences of *K. pneumoniae* strains were analyzed by K- and O-locus (polysaccharide capsule) typing with Kaptive software [33] in comparison to a reference database of known *Klebsiella* capsule types.

### 2.7. Phage Adsorption and One-Step Growth Assay

Percentage of adsorbed bacteriophages was determined using phage adsorption assay with chloroform-mediated inactivation, as described previously [34]. The experiment was repeated three times for both LASTA and SJM3 phages.

Phage latent period and burst size were determined by a one-step growth experiment according to the protocol described by Kropinski [35], with minor modifications. In the adsorption flask, 5 mL of Ni9 log culture (OD_600_ 0.2, 1 × 10^8^ CFU/mL) was infected with 50 µL of phage preparation (10^7^ PFU/mL). After 5 min of adsorption, 1/100 dilution was made by transferring 100 µL of the phage–bacteria mixture to flask A, containing 9.9 mL of fresh medium prewarmed in water bath, and from this moment the time was calculated. Successively, several 1/10 serial dilutions were made in flasks B, C and D by transferring 1 mL of given suspension to 9 mL of fresh medium. At various time points, 100 µL was removed from the appropriate flask, mixed with 15 µL Ni9 overnight culture (2 × 10^9^ CFU/mL) in 10 mL of molten LB top agar and overlayed on LA plates. Plaques were counted after overnight incubation of the plates at 37 °C.

### 2.8. Host Range Determination

The phage lytic activity towards 140 isolates of *K. pneumoniae* from the laboratory collection was tested using spot test [23]: serial dilutions of phage suspensions were spotted onto the surface of solidified LB top agar (10 mL) mixed with 15 µL of overnight culture of tested strains. The appearance of zones of lysis and individual plaques was observed after overnight incubation of plates at 37 °C.

### 2.9. Thermal Inactivation of Phages

The effect of temperature on phage activity was studied by exposing test tubes with 3 mL of phage suspension (1 × 10^8^ PFU/mL) in SM buffer for 10 min to temperatures ranging from 30 °C to 80 °C, by intervals of 10 °C, in a water bath [36]. In the second experiment, the incubation temperatures were narrowed down to a range from 60 °C to 70 °C, with intervals of 1 °C. Extended exposure to high temperatures was stopped by transferring tubes to 20 °C water bath. Phage viability was tested by spot test [23].

### 2.10. The Effect of pH on Phage Stability

Phage cultures were prepared in SM buffer of different pH values (2, 5, 7, 9, 11; adjusted by addition of 5 M NaOH or concentrated HCl) to a final concentration of 1 × 10^4^ PFU/mL and incubated overnight at room temperature [36]. Phage activity and titer were tested by double agar overlay plaque assay [21].

### 2.11. Phage Lytic Activity and Resistant Clone Screening

Interaction dynamics of phage and planktonic bacteria was tested in a 96-well microtiter plate (Tissue Culture Plate; Sarstedt) in triplicate, according to the protocol described by Vukotic et al. [22]. The *K. pneumoniae* Ni9 overnight culture was adjusted to 0.5 McFarland units and further diluted 100 times in 200 μL LB distributed in wells. Phages SJM3 and LASTA were applied separately, as well as in a cocktail suspension (mixture of 50:50), at final MOI 10 (growth control excluded from treatment with phage). At 3, 6, 9 and 24 h after the onset of the experiment, viable cell count was enumerated by spotting dilutions on LA plates [37]. A total of 80 colonies (ten from each time point for each phage) were collected, propagated and analyzed for sensitivity to both phages using spot testing [23]. Among colonies that appeared resistant, eleven were selected for further investigation of phage-resensitization phenomena and designated Ni92–Ni912. They were chosen from each time point and each infection setting.

### 2.12. Antibiofilm Activity

The ability of phages SJM3 and LASTA to lyse host cells in preformed biofilm on porous glass beads was investigated as previously described [22,38] with minor modifications. Glass beads (diameter 4 mm, pore size 60 μm, and surface area of ~60 cm^2^; VitraPor; ROBU, Hattert, Germany) were statically incubated with 100-times-diluted overnight culture of Ni9 for 24 h at 37 °C. Further on, beads were washed three times with saline solution and separated in single wells of a 24-well plate (Tissue Culture Plate; Sarstedt). Each was treated with 10^9^ PFU/mL phage suspension (LASTA, SJM3 or cocktail) at predicted MOI 100 in LB medium for 24 h. After 24 h, treated beads and beads from the untreated control were washed with saline, vortexed in microtubes and, subsequently, sonicated in an ultrasound water bath and vortexed once again in order to detach biofilm-embedded bacteria [39]. CFUs of biofilm before treatment, biofilm after phage treatments and biofilm with fresh medium were counted on LA plates.

### 2.13. Lysogeny Testing

Possible integration of SJM3 and LASTA as prophage in the 11 selected phage-resistant mutants was investigated by PFGE, as mentioned in Section 2.2.

## 3. Results

### 3.1. Bacterial Strains

The first 27 isolates were typed and characterized in a previous study [4], and Ni9 was selected as host for phage isolation in this work based on its carbapenem resistance, colistin resistance and *bla* _NDM-1_ gene presence. The other 113 isolates were found to be resistant to all tested antimicrobials, including carbapenems and colistin, and were classified as pandrug-resistant isolates.

### 3.2. Phage Isolation, Plaque, and Virion Morphology

Two phages named LASTA and SJM3 were isolated from sewage water samples using the homogeneous enrichment method with a *K. pneumoniae* Ni9 strain as the bacterial host. Both produce very similar small (up to Ø1 mm) plaques on the bacterial lawn (Figure 1). Sequential purification, amplification and concentration resulted in purified phage stocks of titer 10^11^ PFU/mL.

Morphologies of virions LASTA and SJM3 appear very similar (Figure 2). Both are typical podoviruses with an isometric capsid (diameter: 67.0 ± 1.8 nm (*n* = 22, LASTA); 68.6 ± 1.6 nm (*n* = 19, SJM3) and a thin collar structure beneath (height: 5.9 ± 0.6 nm (*n* = 15, LASTA, 5.7 ± 0.5 nm (*n* = 12; SJM3); width: 21.0 ± 2.0 nm (*n* = 15; LASTA; 21.6 ± 1.8 nm (*n* = 12; SJM3). Numerous globular appendages forming a characteristic disc structure are attached below the collar (height: 11.7 ± 1.4 nm (*n* = 18; LASTA; 12.3 ± 1.0 nm (*n* = 12, SJM3).

### 3.3. Host Range and Sensitive Strains Analysis

Phages LASTA and SJM3 demonstrated the same host range and lysed 5 (Ni9, GN81, 2060, 2179, GN784) out of 140 tested *K. pneumoniae* isolates. These sensitive isolates were typed by PFGE (Figure 3) and it can be concluded, based on their pulsotypes, that these isolates constitute four strains: Ni9, 2060 and 2179 each constituted individual strains, while GN81 and GN 784 belong to the same strain.

### 3.4. Adsorption Rate and One-Step Growth Curve

After 1 min of incubation, 65% of LASTA virions and 63% of SJM3 virions attached to the bacterial receptors. The amount of adsorbed particles increases with incubation time. About 90% of phage particles LASTA and SJM3 are adsorbed to the surface of host cells within 5 min, which is therefore considered as the adsorption time. After 20 min, 97% and 94% of adsorption is reached for LASTA and SJM3, respectively, (Figure 4).

The replication cycle curve was determined by a one-step growth test. The latent period of both LASTA and SJM3 is 80 min, and their burst sizes (average number of phage particles produced by one infected cell) are 187 ± 37 and 155 ± 34 plaque-forming units per infected cell, respectively. The duration of the rise period—during which phage virions are released from lysed cells—is 60 min for LASTA and 50 min for SJM3, and the total replication cycle lasts 140 and 130 min, respectively (Figure 5).

### 3.5. The Effect of Temperature and pH on Phage Stability

The thermal point of inactivation for the phage LASTA is 68 °C, and for SJM3 it is 69 °C (Figure 6). Both phages were stable at a range of pH 5–11, and inactivated at pH 2 (Figure 7).

### 3.6. Phage Lytic Activity

LASTA and SJM3, as well as cocktail of the two, were tested for their bactericidal efficiency on host cells in planktonic culture. Both phages were independently applied at MOI 10, which was also the final MOI of the cocktail of the two phages mixed in a 50:50 ratio. The bacterial reduction of individual phage treatments appear similar, with the most notable logarithmic drop (2.38 log units for LASTA and 3.81 log units for SJM3, compared to untreated control) at 3 h post-infection. After coping with the initial lytic effect of phages, bacterial cells regain their numbers, reaching the same culture density as the growth control after 24 h of incubation (Figure 8). On the other hand, phages applied as a cocktail achieve the best effect (3.99 log units decrease) on host cells 6 h after infections, but the bacteria regrow further on, reachinggrowth control values.

### 3.7. Antibiofilm Activity

After 3, 6, 9 and 24 h of incubation of preformed biofilm with MOI 100 of each phage separately and also combined, a reduction of CFU count was noticed 3 h after treatment, but with almost no effect on the number of viable cells extracted from biofilm 24 h after incubation with phages (Figure 9). Phage SJM3 and cocktail of the two phages demonstrated the same level of antimicrobial activity on biofilm-embedded cells by decreasing the viable cell count by 2.51 and 2.59 log units, respectively, after 3 h, while the phage LASTA had less success, lowering the levels by 1.56 log in the same time frame.

### 3.8. Resistance Clone Screening

In addition to CFU count, the colonies grown after infection with each bacteriophage were assayed for resistance towards both phages. Ten colonies were collected at each time point (3, 6, 9 and 24 h), cultured overnight and screened by spot test using 10^9^ PFU/mL, 10^7^ PFU/mL and 10^5^ PFU/mL of both phages (Figure 10). Results revealed that the incubation of Ni9 with either LASTA or SJM3 produced colonies that were resistant to both phages. However, this resistance seems to be transient, as colonies with varying levels of sensitivity to phages also appeared. In regards to resistance/renewed sensitivity to phages, collected colonies were either resistant (not lysed by any phage concentration applied), semi-sensitive (lysed only by one or two highest phage preparations) or fully resensitized (lysed by all applied phage concentrations). In general, the frequency and the degree of resensitization become more prominent with time, as at time point 1 (3 h post-infection with either phage) all screened colonies are resistant to both phages. However, significant differences in terms of which phage the colonies developed resistance to were observed at later time points. Regardless of the phage initially applied, the resulting resistance to LASTA remained consistent, as only 9 out of 80 analyzed colonies were semi-sensitive to this phage while others were resistant. On the other hand, medium to full sensitivity towards SJM3 is already regained after 6 h by 40% of all picked colonies, with a similar trend observed in all latter time points for both infections, culminating in 50% of colonies being resensitized to SJM3 after 24 h. Ni93, a resistant mutant that was analyzed in detail further on, emerged 3 h post-infection with SJM3.

### 3.9. Lysogeny Testing

Given the suspiciously long latent period of both phages, and the absence of similar lytic bacteriophages in literature data, their possible integration in the genome of their host was investigated using two different approaches—comparing PFGE pulsotypes of eleven phage-resistant mutants to wild-type Ni9 and sequencing both Ni9 and its phage-resistant derivative Ni93. It was demonstrated that PFGE profiles of resistant mutants do not differ from the profile of the parental strain and, therefore, do not indicate the integration of the prophage (Figure 11). Finally, the alignment of whole-genome sequences of Ni9 and Ni93 confirmed the absence of a LASTA/SJM3 genome sequence from these genomes.

### 3.10. Nucleotide Sequence Accession Number

The genomes of *Klebsiella* phages SJM3 and LASTA have been deposited in the NCBI GenBank database under the accession numbers MT251348.1 and NC_054965.1, respectively. The *K. pneumoniae* strains Ni9 and Ni93 have been deposited in the NCBI GenBank database under the accession numbers JAIZBF000000000 and JAPSFC000000000, respectively.

### 3.11. Phage Genome Analyses

Genomic DNA of phages LASTA and SJM3 were sequenced using the Illumina HiSeq 2500 platform (MicrobesNG). Genomes of both LASTA and SJM3 phages are circular and 62,420 bp long, with a GC content of 56%, 76 protein-coding sequences and one tRNA gene detected. The gene coding for RNA polymerase is not present. These phages are mutually identical in 99.98% of their nucleotide sequence, with only 13 single base pair differences. The gene, nucleotide positions and changes in nucleotide and amino acid levels between LASTA and SJM3 phages are listed in Table 1. The majority of mutations are non-silent (10 out of 13) and half of the amino acid substitutions are located in a single gene, recognized as a T7-like tail fiber gene. Polarity-wise, two of these substitutions occur within the same amino acid classification group (phenylalanine to leucine and serine to asparagine), while three result in a change of polarity/charge: threonine to proline, aspartate to asparagine and glycine to serine. The remaining eight mutations are scattered across the genome: two of them are located in the coding sequence for a tail fiber domain protein, while single mutations occurred in genes encoding endoglucanase E1, portal protein, DNK methyltransferase *Hin*dIII and two hypothetical proteins.

No virulence or resistant genes were found in the phage genomes by scanning with VirulenceFinder v. 2.0 and ResFinder ver. 4.1. The analysis of the phage proteomic tree using the ViPTree online server included 137 *Klebsiella* phages selected from the entire proteomic tree deposited in the Virus-Host DB database (https://www.genome.jp/virushostdb/, accessed on 22 February 2023). LASTA and SJM3 phages were clustered in a distinct group (Figure 12) with only two other phages—SopranoGao (isolated in USA, RefSeq/GenBank accession number MF612073.1) and vB_KpnP_ZX1 (isolated in China, RefSeq/GenBank accession number MW722080.1)—with which they share around 95% identity but with query coverage of 76% and 64%, respectively. Based on a more detailed genome analysis, Li et al. (2022) [40] recently supported the classification of these four phages as a new genus named *Lastavirus*, belonging to the class *Caudoviricetes*. The name of the newly formed genus is accepted by the International Committee on Taxonomy of Viruses [41].

The remaining 24% of LASTA and SJM3 genomes do not share homology with any other phage deposited in Genbank database. The total number of predicted transcription terminators by ARNold is 35 in both phage genomes (Appendix A). In the search for regulatory elements, PHIRE identified 14 sequences related to the consensus sequence CTGGTGGTGCTGTTGCTGGT for both phages (Appendix A).

### 3.12. Bacterial Genome Analyses

Genomic DNA of *K*. *pneumoniae* Ni9 and Ni93 was sequenced using Illumina HiSeq 2500 platform (MicrobesNG). *In silico* analysis using Kaptive software revealed the presence of K15 and O4 capsular loci in both whole-genome sequences of wild-type and mutant *K. pneumoniae* isolates Ni9 and Ni93. The percentage of identity was 99.66% for O and 99.85% for K capsules when compared to reference *Klebsiella* capsule types. The O4 capsular locus is shown to be identical in the Ni9 and Ni93 sequences. However, in the mutant strain Ni93 the *wzc* gene encoding for tyrosine-protein kinase on K15 locus was not detected (Figure 13). Analyzed in Strider, it was determined that the *wzc* gene in Ni93 is modified by the insertion of a 777 bp sequence which corresponds to a micro RNA (AI2786v1_1685.mrna).

In addition, it should be noted that a gene for an integrase was identified in the phage genomes; however, it was deduced to be non-functional, as only a truncated version of the gene is present, encompassing only the first 258 bp, as opposed to the >900 bp of the full gene length.

## 4. Discussion

Bacteriophages are growingly considered as an alternative/adjuvant therapeutic option, especially in cases where antibiotics cannot confer a crucial advantage over bacterial infection. This is evidenced by the proliferation of research articles dealing with the effectiveness and safety of phage application in animal studies, case reports and clinical trials [42,43].

*Klebsiella pneumoniae* is one of the Gram-negative pathogens severely threatening healthcare systems worldwide through the emergence of pandrug-resistant and hypervirulent strains. In the case of difficult-to-treat infections, alternative therapies involving the use of lytic bacteriophages were effectively demonstrated in several cases so far. However, phage therapy is still considered as an experimental approach and requires emergency approval by authorities such as FDA in the USA [44], implying that more data and research are needed to pave the way for integrating phage therapy in modern healthcare strategies.

In this study, we characterized two bacteriophages isolated from Belgrade wastewaters using a pandrug-resistant *K. pneumoniae* strain Ni9. Even though host specificity is considered as one of the most distinct traits of bacteriophages, differentiating them from antibiotics, narrow host range presents an obstacle for efficient phage therapy. Only 5 out of 140 *K. pneumoniae* isolates from our collection are sensitive to phages LASTA and SJM3, which could denote a narrow host range. However, these five sensitive isolates were shown to comprise three different strains, which indicates that strain typing should be done for the rest of our collection in order to correctly ascertain phages’ host range, and this will be the object of further work. Due to their narrow host range, the phages LASTA and SJM3 would be more suitable for use in phage cocktails of increased host range or combined with antibiotics in clinical treatment.

Both phages produce plaques of small diameter, reach about 95% adsorption after 20 min and have large burst size, which is in accordance with an unusually long latent period of 80 min. Although such a long latent period may be strain-dependent, this led us to suspect the nature of the phage’s replication cycle, and the possibility of the phages being temperate was investigated in detail and eventually excluded.

Investigating the lytic activity of SJM3 and LASTA, a significant reduction in host cell numbers in a planktonic culture was achieved by both phages 3 h after infection, and 6 h with the cocktail. After the initial bacterial reduction, regrowth of the host was observed, reaching CFU numbers equal to the uninfected control in 24 h. Similar lytic dynamics have been observed in biofilm, with SJM3 performing better than LASTA in both settings. In our previous work, we perceived an equal phenomenon of host regrowth with *Acinetobacter baumannii* and phage ISTD [22]. In response to the presence of a phage, a population of phage-resistant cells is selected for, overcomes infection and restores the host growth. Although the phage-insensitive population emerges relatively fast, the initial bactericidal efficiency should not be disregarded, especially when considering the use of phages together with antibiotics. Verma et al. [45] have demonstrated that combined therapy consisting of a lytic phage and ciprofloxacin significantly decreased the occurrence of both phage-resistant and ciprofloxacin-resistant strains of *K. pneumoniae* and also had enhanced antibiofilm activity, as compared with individual treatments. Townsend et al. (2021) [46] demonstrated that for a bactericidal score longer than 12 h and successful therapy of *K. pneumoniae*, multiple phages must be applied together. Against host biofilm, the cocktail of LASTA and SJM3 did not demonstrate advantage over the effect of singular phages applied. Although the cocktail was more efficient than the two phages at a later time point (6 h) against planktonic cells, we cannot adjudicate the performance as more potent. As was somewhat expected, combining two very similar phages does not result in obvious improvement of the bactericidal effect; however, we would suggest LASTA and SJM3 as an addition to other familiar *K. pneumoniae* phages, covering a broader spectrum of host strains.

The evolution of phage resistance in host bacteria is frequent and should not be overseen when considering phage therapy. Researching phage-resistant mutants helps discerning a comprehensive combination of phages targeting various host molecules for overcoming all possible resistant variants of the host. Furthermore, as the phage–bacteria co-evolutionary “arms race” implies, the administered phage population might also counter-evolve to circumvent the protection gained by the host, and infect the phage-resistant bacteria [43,47]. This is why the colonies of *K. pneumoniae* Ni9, grown following the phage infection, have been retested for phage sensitivity. Distinctly, all tested colonies from the 3 h time point were phage-resistant, but it came to light that the phenomenon was of a transient nature. A phage-sensitive population re-emerges only 6 h after infection. It is worth pointing out that SJM3 once more proved to be more potent than LASTA, overcoming and lysing previously resistant host cells much sooner. Because of the noted phenomenon—but, more importantly, given a relatively slow replication cycle—we proceeded to investigate if phages LASTA and SJM3 were, in fact, temperate. The total DNA of the eleven selected resistant colonies were subjected to PFGE analysis in order to visualize the possible integration of the phage genome as an additional band or a shift in the PFGE profile, compared to the original strain Ni9. Results did not indicate the integration of LASTA or SJM3 genomes. The final rebuttal of lysogeny was achieved by sequencing a resistant clone Ni93 that showed no sign of prophage insertion. Taken together, we can safely claim that phages LASTA and SJM3 are virulent and this is, to the best of our knowledge, the first report of virulent phages with 80 min of latent period. The presence of a truncated integrase gene in both phage genomes suggests that they might have been temperate in their past but lost this possibility during their evolution. This might explain their prolonged time before inducing the host cell lysis, as this trait did not yet, conditionally speaking, change towards a more virulent phenotype.

One of the most common adaptations of bacteria in response to phage attacks is the development of resistance through preventing phage adsorption by modifying cell surface structures [48]. Therefore, looking for differences in the genomes that would explain the development of phage resistance, we performed in silico analysis of the capsular K- and O- loci of wild-type Ni9 and mutant Ni93 through the Kaptive software. The presence of K15 and O4 loci was revealed, with the O4 remaining intact in the mutant strain Ni93, but with the *wzc* gene, encoding a tyrosine-protein kinase, disrupted on the K15 locus. Namely, Wzc proteins facilitate high-level polymerization of capsular polysaccharides, dictate the chain length, and are an essential component of the regulatory network for capsule synthesis [49]. The differences in the variable regions of *wzc* gene are distinct and correspond to a certain capsular type of *K. pneumoniae.* Pan and collaborators [50] developed a useful capsular typing method through the amplification of the *wzc* region of 78 identified capsular types of *K. pneumoniae,* among which 2 capsular types lacked an amplifiable *wzc* gene and were proven to be acapsular. Moreover, observing the development of mutations conferring phage-resistance in *K. pneumoniae*, Hesse et al. [14] determined the occurrence of mutations in different genes, including *wzc*, with impaired phage adsorption as a result. Hence, we suspect that *wzc* mutation is a plausible molecular mechanism for phage resistance in Ni93.

In addition, bacteriophage genomes were also sequenced. Comparing the whole-genome sequences of SJM3 and LASTA, interestingly, it was observed that more than half (7 of 13) of all mutations were detected in two genes coding for tail fiber proteins (recognized and annotated as tail fiber domain protein and T7-like tail fiber). The bacteriophage tail fiber is strategically positioned at the distal end of the tail and is involved in the binding of the phage to a specific receptor on the surface of host bacteria, such as the capsule, lipopolysaccharides (LPS), transmembrane proteins or even pili or flagella [51]. Consequently, the tail fibers—or, rather, receptor binding proteins (RBP) located on them—have a key role in the determination of host specificity [52]. Studying the tail fiber proteins and understanding their interaction with the host receptor has become increasingly important, with the idea of future engineering of tail fibers for a broader host range [52,53]. Hence, it is not surprising that tail fiber protein genes carry the bulk of mutations. In addition, as previously mentioned, 76% of LASTA and SJM3 genome sequences share 95% identity with phage SopranoGao; however, when it comes to tail fiber protein, this identity is greatly reduced to 31%, as only the first 150 aa (out of ~750 aa) share some level of identity, the rest being non-homologous. Hence, we can assume that this gene is a hot spot for genome evolution as it is involved in host recognition and is pressured in the “arms race”. In the putative tail fiber protein of the here-described phages, the five amino acid substitutions are not localized in the same domain, but are rather scattered in a way that two non-polar amino acids are placed closer to the N terminal and three polar amino acids are placed in the middle and C terminal domain, which was shown to be responsible for receptor binding in T7 [54]. Each of the observed substitutions could have an impact on the 3D conformation of the fiber protein, as it is known that the conformation of the protein arises from the bonding arrangements within its structure. Since the substitutions comprise a total of five changes—including a change of charged amino acids to amino acids bearing no charge, or installing a proline which is known to influence protein-folding by introducing rigid turns into the peptide chains and by setting borders of β-sheets and α-helices [55]—we speculate these changes are responsible for SJM3’s better lytic capacity and temperature stability as well as for the difference in resensitization occurrence. Although LASTA and SJM3 have equal host ranges regardless of the mutations that differentiate them, perhaps the variant of tail fiber proteins of SJM3 forms an efficacious receptor-recognition site that makes phage SJM3 more potent in host elimination. Having all this in mind and observing the bactericidal efficiencies of two phages, we can conclude that, although they differ in only a few base pairs, phage SJM3 is better adapted to the given host bacteria (strain Ni9) than LASTA. Among the remaining eight substitutions, it is important to mention the other two mutations also appearing in another phage-tail-related protein and a single mutation present in a neighboring putative endoglucanase E1 gene. Endoglucanases are one of three key participants in the enzymatic depolymerization of cellulose to glucose [56]. Due to its cellulolytic activity, this enzyme could potentially be involved in the degradation of host EPS. It is peculiar that only two other phages, isolated in very distant parts of the world, share homology to some degree with the herein-described phages and, thus, form a unique genus *Lastavirus*. Excellent insight into genomic characteristics and relatedness of the genus members is given in the work of Li et al. (2022) [40]. Although similar, several differences are striking; ZX1 phage is temperate with three lysogen genes encoding the repressor, anti-repressor, and integrase, but still demonstrates shorter life cycle parameters than LASTA and SJM3. Additionally, it possesses high depolymerization activity, which was not noted in our phages. It would be interesting to see whether host ranges are overlapping, and this along with other characterizations of the *Lastavirus* genus will be the object of our further work. It should also be mentioned that almost a quarter of the LASTA and SJM3 genome does not share homology with any phage. Delving into this region could produce valuable knowledge about these phages. In conclusion, phages LASTA and SJM3 are novel virulent podoviruses with unusually long latent periods and a narrow host range. They are devoid of genetic determinants incompatible with phage applications, including lysogeny, antibiotic resistance, toxicity, transduction and virulence determinants. Although harboring only a few mutations, their performances differ significantly. SJM3 is better adapted to a given host, as it has better bactericidal efficacy both over planktonic and biofilm bacteria, but has also demonstrated lytic activity against a large proportion of bacteria emerging as resistant from the initial contact with both LASTA and SJM3. Regarding their therapeutic potential, more rigorous studies are needed for both LASTA and SJM3 to be considered candidates for application, combined with antibiotics or other phages. A number of challenges still remain in phage therapy, including the right dose of phage, timing of administration, duration of therapy, local regulations, testing of bacteria’s susceptibility to phage in the presence or absence of potential antibiotics as well as the fact that phages directed at a single bacterium cannot eliminate polymicrobial infections.

## Figures and Tables

**Figure 1 viruses-15-00628-f001:**
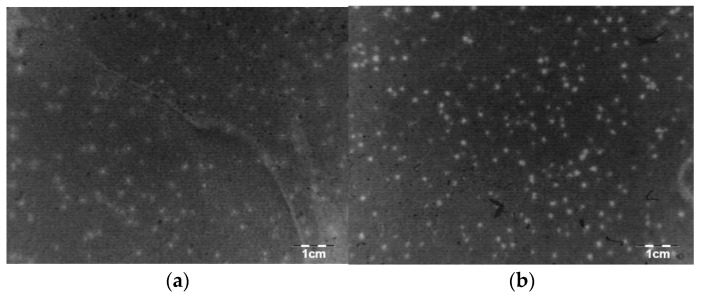
Plaque morphology (**a**) LASTA; (**b**) SJM3.

**Figure 2 viruses-15-00628-f002:**
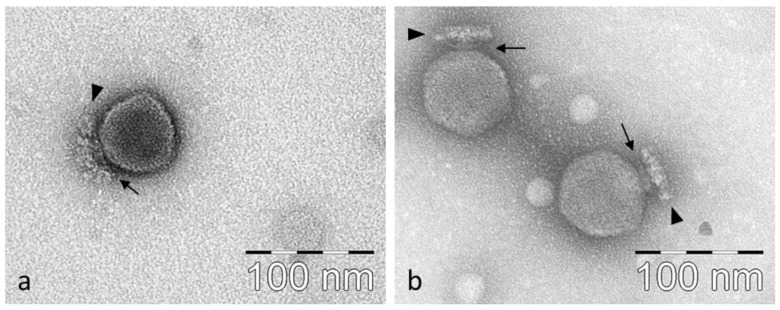
Transmission electron micrographs of *K. pneumoniae* phages LASTA (**a**) and SJM3 (**b**). A thin collar structure beneath the capsids is indicated by the arrows and a distal disc structure composed of numerous globular appendages is labelled with triangles.

**Figure 3 viruses-15-00628-f003:**
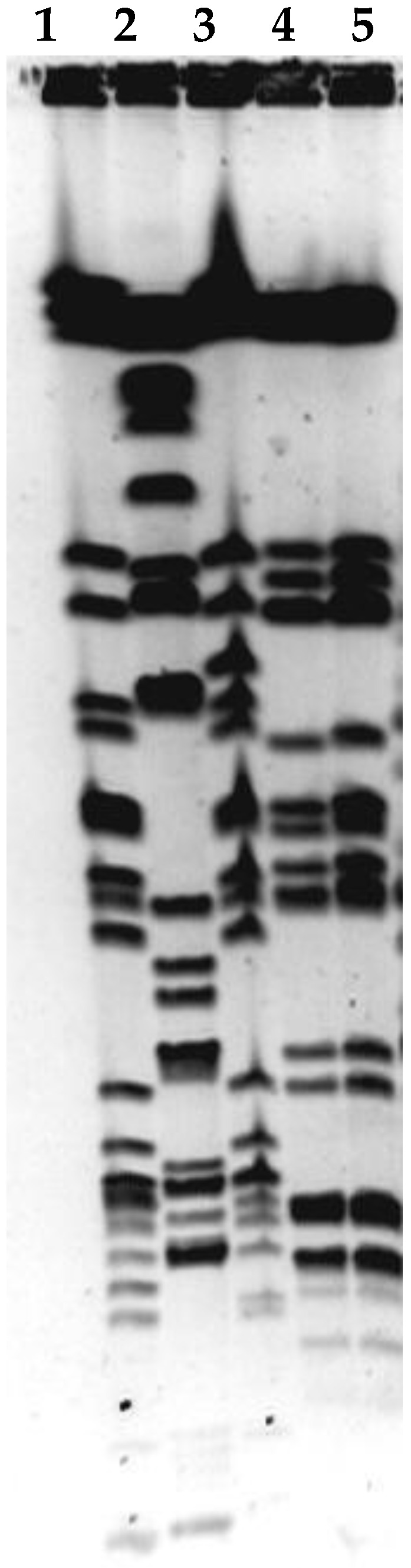
Pulsed-field gel electrophoresis of phage-sensitive isolates. *Xba*I macrorestriction pattern; (**1**) Ni9, (**2**) 2060, (**3**) 2179, (**4**) GN81, (**5**) GN 784.

**Figure 4 viruses-15-00628-f004:**
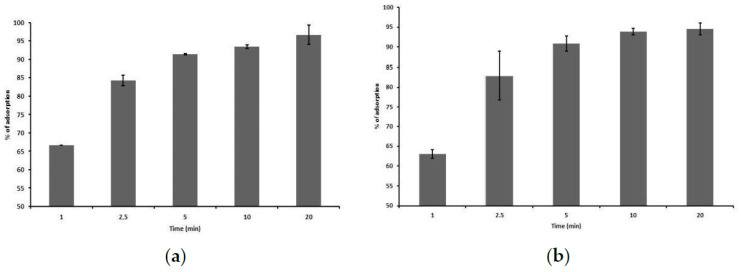
Adsorption rates of phages on *K. pneumoniae* Ni9. Bars indicate the percentage of adsorbed phages at different time points. Each bar represents the mean of three independent measurements. (**a**) LASTA; (**b**) SJM3.

**Figure 5 viruses-15-00628-f005:**
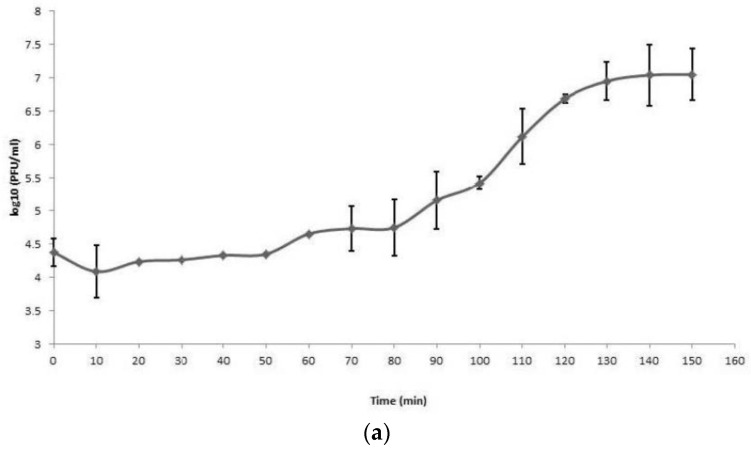
One-step growth curve of phages on *K. pneumoniae* Ni9. Phages were grown in an exponential phase culture of *K. pneumoniae* Ni9. Data points indicate the PFUs/mL at different time points. Each data point represents the mean of three independent measurements. (**a**) LASTA; (**b**) SJM3.

**Figure 6 viruses-15-00628-f006:**
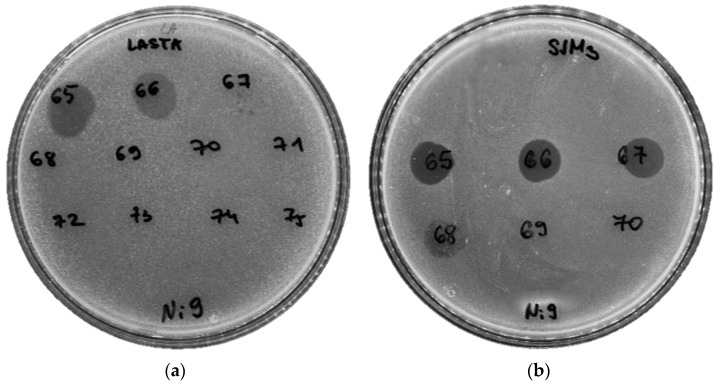
Temperature stability of phages. Thermal point of inactivation for (**a**) LASTA is 68 °C, and for (**b**) SJM3 it is 69 °C.

**Figure 7 viruses-15-00628-f007:**
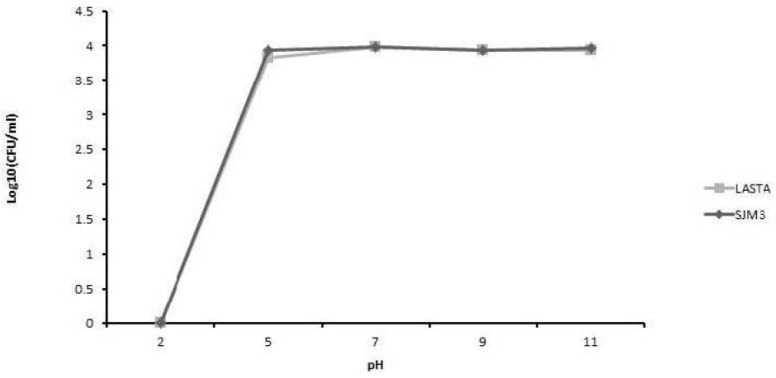
The effect of pH on phage stability. LASTA and SJM3 are inactivated in the extreme acidic environment of pH 2, while they are stable in the range of pH 5 to pH 11.

**Figure 8 viruses-15-00628-f008:**
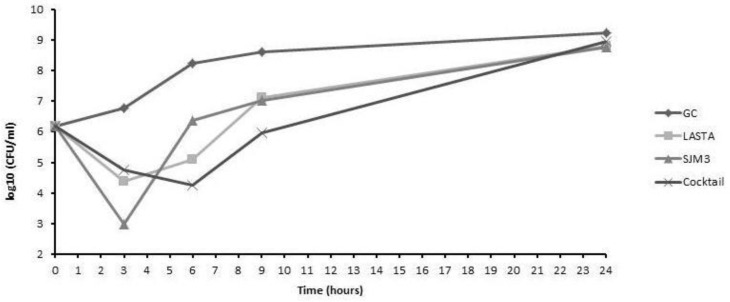
Bacterial cell counts after treatment of planktonic culture of *K. pneumoniae* Ni9 with bacteriophages LASTA, SJM3 and a cocktail of both phages. Independently applied phages achieved the best result after 3 h, yielding a reduction of 2.38 (LASTA) and 3.81 (SJM3) log units in CFU number compared to the untreated control. Phage cocktail performed best after 6 h, lowering CFU count by 3.99 log units. GC: growth control without phages. *N* = 3.

**Figure 9 viruses-15-00628-f009:**
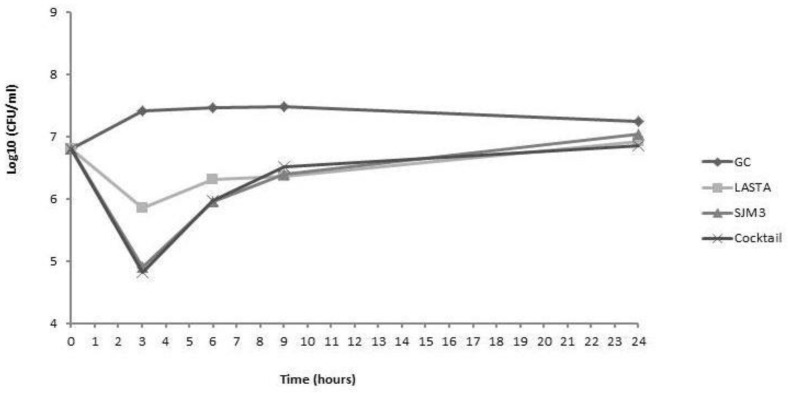
Bacterial cell counts after treatment of biofilm formed by strain Ni9 with bacteriophage LASTA, SJM3 and a cocktail of both. A similar reduction of viable cell count of 2.51 and 2.59 log units was observed 3 h after treatment of biofilm with phage SJM3 and cocktail of the two phages. Phage LASTA produced 1.56 log reduction in the same time, whereas after 24 h, no difference in CFU numbers was observed between treated samples and untreated control. GC, growth control without phages.

**Figure 10 viruses-15-00628-f010:**
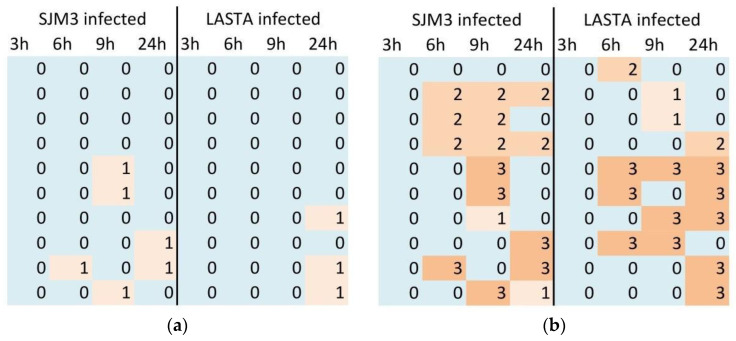
Phage resistance; 0—no lysis zones, 1—zone produced by 10^9^ PFU/mL phage, 2—zones produced by 10^9^ PFU/mL and 10^7^ PFU/mL phage, 3—zones produced by 10^9^ PFU/mL, 10^7^ PFU/mL and 10^5^ PFU/mL phages. (**a**) Resistance to LASTA; (**b**) Resistance to SJM3.

**Figure 11 viruses-15-00628-f011:**
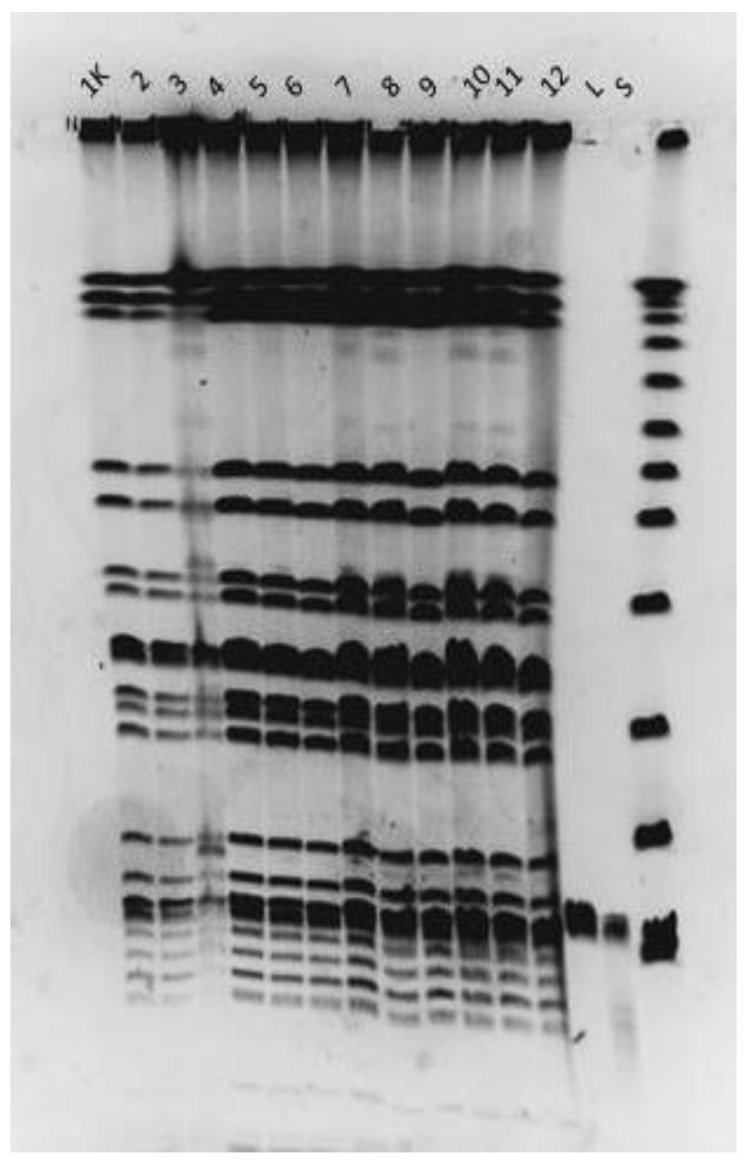
Pulsed-field gel electrophoresis of selected phage-resistant strains. *Xba*I macrorestriction pattern; (**1K**) Ni9 parental strain (**2**–**12**) resistant mutants, (**L**) LASTA, (**S**) SJM3.

**Figure 12 viruses-15-00628-f012:**
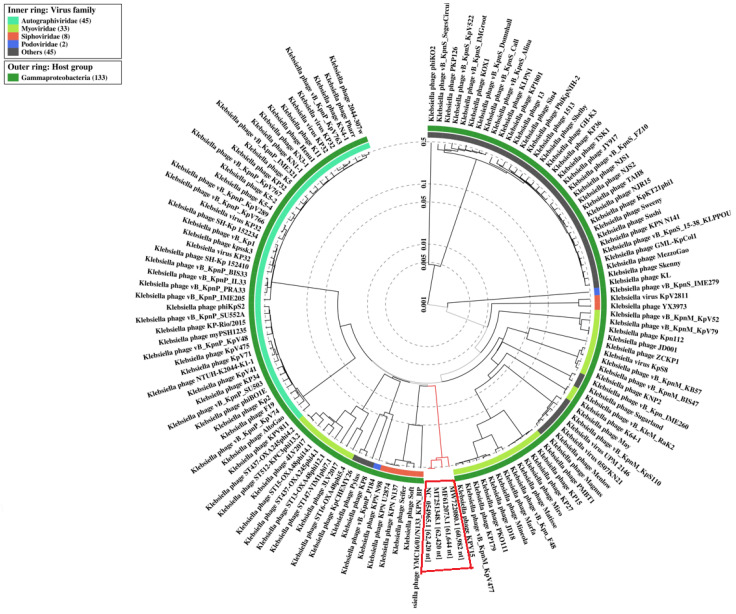
ViPTree Phylogenetic analysis constituted from genome-wide sequence similarities computed by tBLASTx. The circular proteomic tree based on the whole-genome sequences of 137 *Klebsiella* phages was generated by ViPTree software. LASTA and SJM3 phages were clustered in a discrete group with only two other phages—SopranoGao (accession number MF612073.1) and vB_KpnP_ZX1 (accession number MW722080.1)—classified in the genus *Lastavirus*.

**Figure 13 viruses-15-00628-f013:**
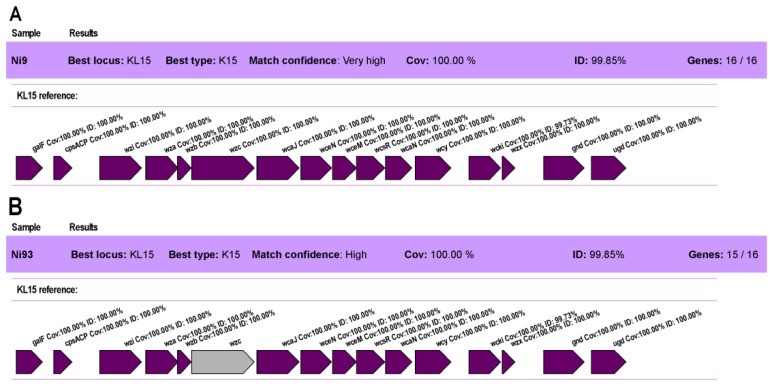
K15 locus analysis with Kaptive software. (**A**) *K. pneumoniae* Ni9; (**B**) *K. pneumoniae* Ni93.

**Table 1 viruses-15-00628-t001:** Nucleotide and amino acid substitutions in the genomes of LASTA and SJM3. Amino acid classification group: green—negative charge, red—non-polar, blue—polar, orange—positive charge.

LASTA vs. SJM3	Nucleotide Substitution	Amino Acid Substitution	Position (bp)	Supposed Function (Based on Protein Alignment)
1	G to A	**E** to**G**	15 299	Tail fiber domain protein
2	T to C	**L**to **F**	15 516	Tail fiber domain protein
3	G to T	**Q**to **K**	23 041	Hypothetical protein
4	C to T	**T**to **P**	24 852	T7-like tail fiber
5	C to T	**F**to **L**	25 119	T7-like tail fiber
6	C to T	**D** to **N**	25 182	T7-like tail fiber
7	A to G	**G**to **S**	25 902	T7-like tail fiber
8	T to G	**S**to **N**	25 996	T7-like tail fiber
9	C to A	**L**to **F**	26 320	Endoglucanase E1
10	C to T	=	35 146	Portal protein
11	A to G	**L**to **P**	46 275	DNK methyltransferase *Hin*dIII
12	T to C	=	51 307	Hypothetical protein
13	T to C	=	53 257	Hypothetical protein

## Data Availability

The genome sequences of phages vB_KpnP_LASTA and vB_KpnP_SJM3 are available from the GenBank database (accession numbers MT251348.1 and NC_054965.1, respectively). The genome sequences of *Klebsiella pneumoniae* strains Ni9 and Ni93 are available from the GenBank database (accession numbers JAIZBF000000000 and JAPSFC000000000, respectively).

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
