# Peer review of "Isolation, Characterization, Genome Analysis and Host Resistance Development of Two Novel Lastavirus Phages Active against Pandrug-Resistant Klebsiella pneumoniae"

_viruses, 2023, doi:10.3390/v15030628_

Round 1
Reviewer 1 Report
This manuscript provides the description and biological characterization of two phages active pandrug resistant Klebsiella pneumoniae
*Numerous similar articles have been published in the recent years and their novelty/significance generally suffers from several faults also found in this study :
- the study includes a very small number of 2 phages which are in addition very close to each other (high genome identity, same host spectrum). This weakens the originality of the authors' findings in the context of phage therapy for which large collections of phages are required to target the diversity of a bacterial species. This phages were isolated from waste water samples using a single bacterial strain, decreasing the chances to isolate various types of phages. Reasons of this choice are not specified. The authors don't comment much on the diversity of anti-Kp phages previously described (taxonomic diversity, activity etc).
- the absence of criteria for selection/genomic characterization of bacterial strains included to assess phage host range. In this study, all strains were carbapenem resistance. Emergence of carbapenem resistance is associated to some successful clones and diversity in this collection may be quite low.
*Authors conclude that the phages could be candidates for phage therapy. This statement should be tempered by their restricted host range and uncertain lysogenic cycles. Indeed, the authors indicate that they identified an incomplete integrase but don't provide informations on the identification of others genes evocating a possible lysogenic lifestyle (repressor, recombinase etc.). In addition, the experiments performed to refute this hypothesis seem insufficient. Indeed, lysogenisation can be a rare event. The authors only screened 11 bacterial clones for phage integration in the bacterial genome.
*Authors also conclude that the long latent period is an original finding. However, this parameter has been evaluated using only one strain but can be strain dependant.
Minor comments :
- the genome analyses section should be divided in two parts : phage and bacteria genomes analyses Phage genome characterization could be improved by taxonomic analyses. Phage classification doesn't rely on morphological types anymore.
Author Response
We wish to express our gratitude to the reviewer for valuable remarks and suggestions that have improved the quality of the manuscript. For detailed response to the reviewer's comments, please see the attachment.

Author Response

(The authors gave the same response as above.)

Reviewer 3 Report
This manuscript displays Isolation, characterization, genome analysis and host resistance development of two new K. pneumoniae phages. The paper is done in a consequential way.
The paper covers the methods and corresponding experiments performed for phage characterization in regards of clinical usage. The level of experiments done could be evaluated as an average, especially ones related to phage lytic activity study. The bacterial clinical isolates, used for phage screening (basic lytic activity study) are not studied to be grouped based on capsule type, receptors etc.; Phage EOPs on the sensitive bacteria (totally 3.6%) are not calculated; bacterial isolates are not present pure isolated bacterial strains, different range of MOIs were not used for phage lytic activity study. So, to make conclusion on lytic activity of the given phages, is not enough. The positive controls in lysogeny, adsorption rate test and OSG curve test need to be included to validate the experiments.
If genome sequencing is enough requirement/criterium, It could be published just as a kind of report.
I would give some suggestions/comments to be addressed:
Page 19. “their virulent nature was confirmed” – better to write lytic nature and further on as well. Or define “virulent”
Page 22. “demonstrated significant bactericidal capacity in a time-dependent” -
Results don’t show bactericidal effect, otherwise define and refer the definition.
Page 66. “may contribute significantly to the pathogenicity” -
Better to write increases the degree of pathogenicity, as Pathogenicity is the capacity of an organism to cause disease.
Klebsiella pneumoniae is a medically important pathogen that produces a thick protective capsule that is essential for pathogenicity.
The thick CPS layer of K. pneumoniae is essential for virulence and forms a protective barrier surrounding the bacterial cell
https://doi.org/10.1128/ Spectrum.01023-21.
Page 86. “Bacterial identification was done by routine microbiological identification using Vitek 2, MALDI-TOF, and API 20E” - Remove Bacterial identification
Pages 88-89.
“All isolates were found to be resistant to all tested antimicrobials, including carbapenems and colistin and were classified as pandrug-resistant isolates”- Remove this as it be in the section of results.
Pages 88-89.
“K. pneumoniae Ni9 strain [4] was further on used as a host bacterium for bacteriophage isolation and propagation.” - Why it was selected as a propagation host strain could be explained in the results.
Pages 91-92.
- “All bacterial strains were propagated in Luria–Bertani (LB) medium” -add company/trade name.
And indicate that the same media was used of phage propagation and testing.
- “and were stored in LB supplemented with 15% (vol/vol) glycerol at −80°C”- Remove this as it is used for bacterial culture maintenance and not particularly for this work.
Pages 93-94.
- “Titer of phages was determined with plaque assay, by mixing 100 μl of phage dilution – write pfu/ml of it “and 15 μl” – write OD and corresponding cfu/ml of it.
- “SM buffer (100 mM NaCl, 8 mM MgCl2 × 7H2O, 50 mM Tris-HCl pH 7.5)” – indicate if it was laboratory made or aif not add company/trade name.
- “of overnight culture of Ni9 strain in 10 ml of melted LB top agar (0.75% agar) and pouring over a thin layer of 1.5% LB agar in Petri dish” - Name the method as there is different “plaque assay”, and refer. Highlight what did you modify in and why. for example, volume and percentage of top agar as well.
- “thin layer of 1.5%” – it is call solid agar layer. Usually, it is not thinner than top agar.
Pages 106-108.
Indicate that it was homogeneous enrichment method and was enriched just with one bacterial strain and not the matrix of different ones.
Pages 108-109.
- “sewage water filtrate was mixed with 2X concentrated LB medium and with overnight culture of host strain Ni9.” - This sentence doesn’t say any significance if you don’t indicate the final cfu/ml (or OD and volume of bacterial suspension used and its corresponding cfu/ml) of used enrichment strain.
- “After 16 h incubation” – write the temperature used as well
Page 110.
- “collected supernatant of the mixture” – it is better to write supernatant of potential lysate instead of mixture; and indicate if it was filtrate or directly tested without filtration.
- “was tested for phage presence by plaque assay” -again name the method and refer.
Page 111.
”purified from samples through four rounds of single plaque purification” – name the method and refer.
Page 111-112.
- “Final single plaque was inoculated into mid- exponential-phase culture of strain Ni9 for overnight propagation and preparation of a working phage suspension, used in further tests.” – it would be clearer to write that the single plaque from last round of plaque purification was used for production of working phage lysate.
- Refer the phage propagation method. Why it was selected, does it give high titer, what was the phage/bacteria ration of inoculum to be propagated.
Pages 113-122.
- Refer the phage purification/concentration method
- “In order to obtain a highly purified and concentrated phage suspension, ultracentrifugation in gradient of CsCl was performed” – I would avoid to write “highly” purified as it isn’t further shown if the level of purification was studied/assayed and full criteria of purification identified.
Instead would write- phage culture purification and concentration was performed using….
- “Mid-exponential phase Ni9 culture of 500 ml” - write OD and corresponding cfu/ml.
- was infected with 1 ml of phage suspension (titer 109 PFU/ml)
- Supernatant of the mixturelysate was collected after centrifugation
- finally decontaminated by passing through (the term of decontamination doesn’t fully fit here) filtered using 0,22 μm filters
Pages 127-133.
Purified Phage culture purified by CsCl gradient centrifugation (this is main but still just one step of purification) were dialyzed…
Page 130.
“lysate (100 μL)” – write pfu/ml
Page 138-139.
- Write bacterial DNA isolation method as well.
- “of highly purified and concentrated” - Purified and concentrated Phage culture of (write volume and pfu/ml)
Pages 140-141.
Refer the method
Page 146.
“Total genomic DNA of K. pneumoniae Ni9 and its phage-resistant clone Ni93, as well as genomes of phages” - better to write just phage and bacterial genomic… as naming phages, bacterial strains and their amount that was sequenced should go to section of results.
Page 160.
The same
Pages 164-169.
Whole section should go to results or write in generic manner.
Pages 173-174.
“To determine the number of phages that attach to the cell surface of host bacterium during a certain time period, an adsorption test was performed. Percentage of adsorbed bacteriophages were determined as described previously [23, 16].”
Instead - Phage adsorption rate was determined using the phage adsorption assay as described previously [23, 16]. And name the method as there are two main adsorption method using different neutralizing means.
Pages 176-177.
Instead - Phage latent period and burst size were determined by a one-step growth experiment according to protocol by Kropinski [24], with minor modifications.
Pages 178-181.
- “in the adsorption flask, 5 ml of Ni9 log culture” – write OD and corresponding cfu/ml
- “After 5 minutes of adsorption” – based on your results, adsorption took totally 20 min.
- “protocol by Kropinski [24], with minor modifications” - Particularly?
- “at various time points, 100 μl was removed” -based on classical OSG method, samples for pfu/ml are taken in every two minutes.
Pages 179-182.
- “1/100 dilution was made by transferring 100 μl of the suspension to flask A, 179 containing 9,9 ml of fresh prewarmed medium, and successively several 1/10 serial dilutions were made in flasks B, C and D, by transferring 1 ml of given suspension to 9 ml of fresh medium. At various time points, 100 μl was removed 181 from the appropriate flask”
- this is very confusing. Just write, that 1/10 and 1/100 dilutions of samples from the experimental flask were made correspondingly at various time points and targeted dilutions were subjected to phage enumeration (determination of pfu/mL) method or plaque assay and refer the method already described above in the paper body.
- “prewarmed medium” – usually the experiment is done using water bath for all tubes with contents employed.
mixed with 15 μl Ni9 overnight culture in molten LB top agar, and overlayed on thin LB 182 agar plates. Plaques were counted after overnight incubation of the plates at 37°C.
Page 185.
Host range determination
Page 187-189.
- “The ability of phages SJM3 and LASTA to lyse 140 isolates of K. pneumoniae from laboratory collection was inspected using spot assay”- instead - the phage lytic activity towards…. was tested using spot assay and please refer the method.
- “Serial dilutions of phage suspensions were spotted onto the surface of solidified LB top agar mixed with 15 μl of overnight culture of tested strains. Appearance of zones of lysis was observed after overnight incubation of plates at 37°C.” – if serial dilutions was tested then not just clear zones, but individual plaques must be counted to pfu/ml.
Page 192.
“Killing dynamics and resistant clone screening”
- Better to write bacterial reduction curve or phage lytic activity dynamics as “Killing” is not necessary to produce reduction curve.
Pages 196-198.
- Add reference of the method
- The sentence needs to be rephrased, like: Phages SJM3 and LASTA were applied separately and in a cocktail (mixture of 50:50) as well at final MOI 10.
- “At 3, 6, 9 197 and 24 h after the onset of the experiment, viable cell count was enumerated by spotting dilutions on LA plates” – define “LA” and add reference for the method (cfu/ml determination).
Pages 199-205.
needs to be rephrased in a shorter and clearer way.
- “Formed colonies were collected and further tested for resistance against the phage they were infected with, as well as against 199 the other phage.
- Ten colonies grown from each time point were picked” – is it the same colonies mentioned above?
- transferred into LB broth, grown overnight and plated in 10 ml soft agar, followed by spotting both phages in concentrations of 109 PFU/ml, 107 PFU/ml and 105 PFU/ml – if it is the same method described above? Just name lytic activity determination method and refer.
- Eleven of the resistant colonies – it is confusing. 10 colonies were tested, were not?
- Ni93, a resistant mutant that was analyzed in detail, emerged 3 h post-infection with SJM3 – describe the analyzing method or move this sentence to the results.
Page 212.
“Each was treated with phage suspension (LASTA, SJM3 or cocktail) at MOI 100 in LB medium for 24 h” – how was MOI calculated/ adjusted? Or was it calculated later, once got results from “CFUs of biofilm before treatment”? and explain why was chosen MOI100 in this case.
Pages 220-228.
- “Possible integration of SJM3 and LASTA as prophage in 11 selected phage-resistant mutants was also investigated by PFGE” - define abbreviation as it appears first here. If you don’t give method description/reference, then it must goes just in results explaining importance of it in that aspect.
- “In addition, induction of hypothetical prophage from Ni93 was performed as previously described. Briefly, overnight cultures of K. pneumoniae Ni9 and Ni93..” – it is confusing. Describe just method and name bacterial strains including mutants in results. Or was the experiment done just on two host strains?
- “applied as potential phage source in spot assays” – refer the method
- “using both strains as indicators/hosts for potentially induced phages.”- do you mean as a controls?
- “The following day, petri dishes were inspected for zones of clearing” – this is part of “plaque assay, must not be repeated here if it is the same method described above in the body of paper.
- “Finally, mutant Ni93 was submitted to whole-genome sequencing” – remove this sentence as it doesn’t give description of method.
- Usually any isogeny test must consist of positive control of lysogenic bacterial strain to validate the experiment.
Page 236.
- “Sequential purification, amplification and concentration resulted in highly purified phage” – remove “highly”.
Page 246.
“both phages belong to the order Caudovirales, familiy Podoviridae “– New classification must be used: Podoviruses, class Caudoviricetes
https://doi.org/10.3390/v13030506
Pages 258-261.
“it can be concluded, based on their pulsotypes, that these isolates constitute three strains: Ni9 and 2179 belonging to first, GN81 and GN 784 to second – sharing a large proportion of bands, while 2060 presents a quite distinct third strain.” – the sentence is confusing, needs to be rephrased.
Pages 269-271.
“Within 5 minutes of incubation, about 90% of phage particles LASTA and SJM3 are adsorbed to the surface of host cells. After 1 minute 65% of LASTA virions and 63% of SJM3 virions attach to the bacterial receptors. The adsorption rate increases with incubation time, reaching 97% and 94% for LASTA and SJM3, respectively, after 20 minutes” – it doesn’t sound correct, 90% was adsorbed in 5 minutes and the rest 4-7% in 15 minutes.
Usually – positive control of known phage with known adsorption rate must be used in parallel to validate the experiment performed.
Pages 279-283.
“The latent period of both LASTA and SJM3 is 80 minutes” – does it includes adsorption time as well? If so, then replication would be longer than:
“is 60 minutes for LASTA and 50 minutes for SJM3, and the total replication cycle lasts 140 and 130 minutes, respectively”.
Page 294.
3.4. Bactericidal kinetics – change the bactericidal
Page 298.
“The killing dynamics of individual phage treatments appear similar” – remove or change “killing”
Page 299.
“CFU drop” – logarithmic drop
Page 313.
“with MOI 100 of each phage separately and also” – again, explain why was applied MOI 100 comparing to planktonic treatment.
Page 316-317.
“biofilm-embedded cells by decreasing the viable cell count by 2.51 and 2.59 log units, respectively, after 3 h, while phage LASTA had lesser success, lowering the levels by” – as there is now big difference of phage lytic activity between planktonic and biofilm treatment, that means the degree of biofilm formation is not enough.
Page 330.
“by spot test”- keep using the same name of method. Or refer if this is different.
Pages 330-331.
“The phages were applied in 3 different concentrations (109 PFU/ml, 107 PFU/ml and 105 PFU/ml), which all formed lysis zones on the bacterial lawn of wild type (Ni9) strain” – this is already written in the methods. And define “lysis zone”, if you mean “clear zone” then it doesn‘t prove phage activity, if there is no individual plaques produced. so sentence needs to be rephrased.
Page 333-337.
If different dilutions of phages were used for study of phage lytic activity, then calculate EOP for each strain and evaluate it quantitatively instead of qualitatively. It is difficult to interpret. It doesn’t sound correct: “low-sensitive colonies, medium-sensitive colonies……the resulting resistance to LASTA remained consistent as only a low level of sensitivity”.
Pages 362-363.
“Finally, the alignment of whole-genome sequences of Ni9 and Ni93 confirmed the absence of LASTA/SJM3 genome sequence inside the Ni93 genome.” – why just Ni93 and not Ni9?
Page 428.
“order Caudovirales, and Podoviridae family” – use current new classification.
Pages 431-432.
“Only five out of 140 K. pneumoniae isolates from our collection are sensitive to phages LASTA and SJM3, which could be denoted as a narrow host range” – bacterial strains used for phage screening are not grouped based on capsule types, different receptors or some other genetic characteristics, to make the full conclusion on the host range.
Page 432.
“These five sensitive isolates were shown to comprise three different strains” – does it mean that mixed isolates were used for phages lytic activity study?
Pages 439-448.
- Change bactericidal and killing activity.
- “activity, regrowth of the host is observed, reaching equal CFU numbers as the uninfected control in 24 hours. Similar” – regrowth needs to be defined every time you are using it in whole body of paper, does it consider both re-growth of initial host strain and over-growth of phage-resistant mutants?
Pages 520-521.
“but has also demonstrated lytic activity against a large proportion of bacteria surviving the initial contact with both LASTA and SJM3” – sentence is not clear.
Author Response
We wish to express our gratitude to the reviewer for detailed inspection of our work and valuable remarks that have improved the accuracy of writing and the overall quality of the manuscript. For detailed response to the reviewer's comments, please see the attachment.

Round 2
Reviewer 1 Report
I do not have any additional comments
Author Response
We express our gratitude to the reviewer for investing time and effort in improving the quality of our manuscript. We are thankful for the kind suggestions and corrections.
Reviewer 2 Report
Thanks for the corrections
Author Response

(The authors gave the same response as above.)

Reviewer 3 Report
Page 117: “homogeneous culture of K. pneumoniae Ni9 strain” – when you name bacterial strain culture used for experiments, it means it is pure homogeneous culture of one strain. Previously I meant homogenous (using one strain or different strains of one species) and heterogenous enrichment methods.
Pages 187-202:
- “Phage adsorption rate was determined using phage adsorption assay with chloroform - mediated inactivation, “- here, in this work further you are giving phage adsorption percentage not rate that calculates with formula (K = 2.3 /Bt X log 10 (P0/Pt)).
- “as described previously [3423, 16]” – should be reference number 22? (Vukotic, ….)?
- “In the adsorption flask, 5 ml of Ni9 log culture (OD600 0.2, 1 x 108 CFU/ml)” - this is titer (bacterial titer) as well….“ was infected with 50 μl of phage preparation (titer 107 PFU/ml)” – so here titer is not something different, better to remove.
- “100 μl of the suspension” – it is mixture of phage-bacteria or content of the tube.
- “100 μl of the suspension to flask A, containing 9,9 ml of fresh prewarmed medium prewarmed in water bath, and successively several 1/10 serial dilutions were made in flasks B, C and D” – first of all 100 μl into 9.9 ml is 1/100 dilution not 1/10 and it is not 1/100 serial dilution, it is just one dilution of 1/100. If it is according to Kropinski.
- Why did you use the different of flasks B, C and D if you treat them in the same manner further? At least it is not described.
Pages 213-217:
“The effect of temperature on phage activity was studied by exposing test tubes with 3 ml of phage suspension (1 x 108 PFU/ml)” – refer the method and why this 1 x 108 PFU/ml was selected to be studied for temperature exposition. And why just for 10 min? maybe explain in the results or discussions
“Phage viability was tested by spot test [23]” - The same reference is given here (but one dilution spot is used as it is shown on the picture of the results section, that doesn’t give phage titer reduction results) as for host range determination where spotting of serial dilutions are described – confusing.
Pages 213-217:
“final concentration of 1 x 104 PFU/ml and incubated overnight at room temperature. – again, refer the method and why this 1 x 104 PFU/ml was selected to be studied for temperature exposition. Why was the different phage concentrations used for phage/sensitivity or inactivation experiments?
Page 232: “viable cell count was enumerated by spotting dilutions on LA plates [36].” This reference of 36 (Ben-David..) doesn’t correspond to evaluation method used here: Page 391 and Figure 10 “Resistance clone screening”. And if spotting of dilutions were used, why are not calculating cfu/ml and corresponding reduction instead of qualitative evaluation given in the figure 10? At the same, while working with phages, and phage=bacterial mixture is used for calculation of cfu/ml, sample dilutions afre very important to obtain reliable data.
Pages 279-280: “After enrichment of phages from sewage water samples using Klebsiella pneumoniae Ni9 strain as the bacterial host” – needs to be rephrased: two phages named LASTA and SJM3 were isolated from sewage water samples using homogeneous enrichment method
Page 280: Better write: Both produce very similar small with up to Ø1 mm plaques”
Better write: Stocks with titer of 1011 PFU/ml.
Pages 317-331:
- “Within 5 minutes of incubation, about 90% of phage particles LASTA and SJM3 are adsorbed to the surface of host cells. After 1 minute 65% of LASTA virions and 63% of SJM3 virions attach to the bacterial receptors.” – it is confusing sentence, first 5 minute then plus one min. It is better to write in order that in one minute 65% ..in 5 minutes 90% of particles adsorbs to the host cells surfaces and in 20 minutes the number of adsorbed particles reaches to 97…%. It is better to state adsorption time 5 min and 90% - as phage characteristics. Although in the section of methods for OSG you are considering 5 min as an adsorbing time:
Page 196: “After 5 minutes of adsorption, 1/100 dilution was made by transferring 100 μl of the suspension to flask A”. so how the timing was calculating is not clear…..Does the OSG figures show date obtained after adsorption period?
- “The adsorption rate increases with incubation time” – should be adsorbed particles amount, or otherwise calculate it adsorption rate.
Pages 518-519:
Better to rephrase: “Due to their narrow host range, phages LASTA and SJM3 would be more suitable to use in phage cocktails to increase the host range for clinical treatment.
Author Response

(The authors gave the same response as above.)
